# Understanding the Spatial-Temporal Changes of Oasis Farmland in the Tarim River Basin from the Perspective of Agricultural Water Footprint

**Aihua Long** [1,2,†], **Jiawen Yu** [1,*,†], **Xiaoya Deng** [2,*], **Xinlin He** [1], **Haifeng Gao** [3], **Ji Zhang** [1,2], **Cai Ren** [1,2] **and Jie Du** [4]

1   College of Water and Architectural Engineering, Shihezi University, Shihezi 832000, China; ahlong@iwhr.com (A.L.); hexinlin2002@163.com (X.H.); 18813072717@163.com (J.Z.); cyrus1837@163.com (C.R.)
2   State Key Laboratory of Simulation and Regulation of Water Cycle in River Basin, Department of Water Resources, China Institute of Water Resources and Hydropower Research, Beijing 100038, China
3   Ministry of Ecology and Environment Center for Satellite Application on Ecology and Environment, Beijing 100094, China; gaohf03@hotmail.com
4   Power China Chengdu Engineering Co., Ltd., Chengdu 610072, China; taiyuandujie1992@163.com
*   Correspondence: yujiawen_415@163.com (J.Y.); lily80876@163.com (X.D.)
†   These authors contributed equally to this study and share first authorship.

**Abstract:** The Tarim River Basin in China has predominantly assumed the task of commodity cotton and other high water-intensive crop production in recent years. The spatial matching status of agricultural water and land resources is a prerequisite for local economic development. This paper provides an insight into the spatiotemporal variation trends of agricultural production water footprint and oasis farmland in the Tarim River Basin. The degree of spatial mismatching between oasis farmland and crop production water footprints studied in this paper found how the crop water footprint affected the change in oasis farmland area by sensitivity analysis. Time series data covering the period of 1990–2015 were used for the study. The results showed that the annual variation of crop production water footprint and oasis farmland area have experienced upward trends in Tarim River Basin. The blue water makes the largest contribution to the components of the crop production water footprint in each district (all exceeded 77%). The crop production water footprint and oasis farmland area tend to aggregate towards the eastern region. The level of spatial mismatch between the blue water footprint and farmland area fluctuated during the study period, but it was gradually remedied after 2000, while the spatial mismatch between green water footprint and farmland area gradually worsened. The number of districts with mid and high sensitivity to changes in blue water footprint continuously increased during 1990–2005, which revealed that the change in blue water footprint has an increasing influence on oasis farmland. The results can provide operable recommendations for efficient use of water resources, maintaining oasis suitable farmland scale and agricultural sustainable development in the Tarim River Basin.

**Keywords:** oasis farmland; crop production water footprint; Standard Deviation Ellipse; Gini coefficient; spatial mismatch index

## 1. Introduction

The increasingly global scale of environmental degradation in the Anthropocene has engendered broad discussion about the conflicts between water and land resources in inland arid oases [1]. Shortages and inconsistencies between available water and farmland resources are long-term and radically restrictive factors in the process of agriculture modernization [2]. The Tarim River Basin (TRB) is the largest inland basin in China, and within the background of the strategy Western Development Program, the TRB government

has undertaken the task of commodity cotton production [3]. This task involves a total cotton yield accounting for more than 48.4% of the production in China. Especially in the past 25 years, because of the impact of high-intensity agricultural exploitation, the newly added farmland area was $12.94 \times 10^3$ km$^2$ in TRB, and about 88.7% of the newly added area has been converted from natural grassland [4]. In addition, irrigation water has long-term accounted for more than 90% of the total gross requirements. Soil erosion and pollution caused by the large-scale occupation of agricultural water and land resources have caused the quality of the wetland environment to degrade in the oasis [5]. The volume and reasonable utilization or not of water both directly impact the production efficiency and utilization pattern of oasis farmland, while the degree of reclamation of oasis farmland restricts the development and utilization of water resources [6]. Therefore, making spatial-temporal changes to farmland and agricultural water, and the matching characteristics research between them, is of great practical significance for the optimal allocation of oasis agricultural water and land resources to sustain agricultural production capacity.

Thus far, multiple studies on arid regions have focused on the spatial-temporal distribution characteristics and the matching of water and land resources. Li et al. (2016) evaluated this situation concerning agricultural water and land resources in Yan'an city of China by constructing Lorenz curve and calculating agricultural water and land resource mismatch model [7]. Their research found that the mismatch degree between agricultural water and land resources was higher than that at the national average level for the same period. Obvious spatial differences were observed in the degree of mismatching, which generally improved from south to north. Du et al. (2019) analyzed the matching degree between irrigation water and farmland resources in Ningxia Province of China from the perspective of a Gravity Center model. Results showed that the gravity centers of irrigation water and effective irrigated area both moved in the southeast direction, and their matching degree was superior to that between irrigation water and farmland area [8]. Zhang et al. (2020) studied the matching pattern between renewable water resources and farmland area in Central Asia by constructing a Gini Coefficient model. The authors concluded that the large spatial differences in the matching degree in water distribution and utilization among Central Asian countries, along with the overexploitation of land resources, have ultimately led to the serious water crisis [9]. As shown above, the spatial-temporal distribution and matching situation between water and land resources has been mainly calculated using the following three methods: (i) studying the equilibrium status of regional water and land resources by drawing a Lorenz curve and building Gini Coefficient model, (ii) investigating the spatial–temporal change trend of water and land resources based on the Gravity Center model, and (iii) based on the statistical yearbook's water consumption and land area data, analyzing the matching index of water and land resources using the water consumption per unit land area as the measurement indicator. Although the aforementioned methods are somewhat successful for measuring the spatial-temporal changes and matching of water and land resources, certain shortcomings have been observed: (i) the Lorenz curve and Gini Coefficient can represent only the overall matching degree between water and land resources, and it reflects the relative relationship between each sub-region in the study area but cannot reveal the absolute matching situation; (ii) research on the spatial-temporal changes of water and land resources include the distance of the center of gravity shift as well as the direction distribution, spatial expansion and aggregation, and thus, the center of gravity model provides unilateral and incomplete results when the center of gravity shift of water and land resources are used only as the explanatory variables; (iii) the matching index can display the relative spatiotemporal ratio between water and land resources, and the land area parameters obtained from the statistical yearbook cannot directly express the specific geographical spatial images. The water resource parameters mainly included water resource volume [10], available water resource volume [11] and irrigation water volume [12]. Currently, a paradigm shift has been observed in the field of global water consumption, with the core element represented by the production water footprint theory [13]. The introduction of production water footprint theory provides new

methods and ideas to analyze the real demand and occupation of oases to agricultural water resources from the perspective of socio-hydrology. It can accurately reflect the use of water resources in agricultural production [14]. Veettil and Mishra (2016) considered that, broadly defined, water consumption in agricultural production (crop production water footprint) can be divided into two parts: blue water footprint and green water footprint. The blue water footprint refers to the consumption of surface water and groundwater. The green water footprint refers to the consumption of rainwater insofar as it does not become runoff [15]. Hoekstra and Mekonnen (2012) estimated that the water footprint used for global agricultural production is approximately 7106 $Gm^3/a$, including blue water footprint at 920 $Gm^3/a$ and green water footprint at 6186 $Gm^3/a$ [16]. The research results of Zhang et al. (2018) indicated that more than 90% of the agricultural production water footprint was dependent on the blue water footprint in Xinjiang Province [17]. Thus, the water resource volume, available water resource volume and irrigation water volume are all unilaterally used as characteristic parameters of the water resource consumption of a region, and the matching situation should be considered from the perspective of the blue–green production water footprint.

This paper investigates the spatial-temporal changes and matching degree of agricultural water footprint and oasis farmland resources. The present study has the following objectives: (a) to reveal the recent spatial-temporal change trends of oasis farmland resources and crop production water footprint, (b) to construct the measuring model of the spatial mismatching index and Gini coefficient between the oasis farmland resources and crop production water footprint, (c) to analyze the impacts of crop production water footprint change on oasis farmland area, and (d) to establish sustainable strategies to increase the efficiency of blue-green water use and ensure suitable oasis farmland scale. TRB is used as a case study, and the analysis is based on a 25-year period from 1990 to 2015. The results can offer novel thoughts for the sustainable management of oasis farmland and for drafting future agricultural development policies.

## 2. Study Area, Data and Methodology

### 2.1. Study Site Description

The Tarim River Basin (TRB) in northwest China comprises the catchment areas of 9 stream sources and the main stream of the Tarim River. The basin geographical boundary lies between 71°39′ E and 93°45′ E longitude and 34°20′ N to 43°39′ N latitudes. The TRB is approximately 1.02 million $km^2$, covering five districts, namely Aksu, Bazhou, Kashgar, Hotan and Kezhou (Figure 1). Among all the districts, Kashgar makes the largest contribution (~35.8%) to the water footprint of crop production, and Kezhou makes the smallest contribution (~2.4%). The studied catchment experiences an arid-continental climate with an annual average temperature of 10.6~11.5 °C and 2400~3250 h of sunshine. Mean annual evaporation in the watershed ranges from around 1500 to 2500 mm, far exceeding the mean annual precipitation (<80 mm). TRB's artificial oasis area accounts for approximately 4.2% of the basin area. About 96.8% of the TRB's freshwater diversions were for agricultural irrigation. Of the water diverted for irrigation, most (~87.5%) was taken from surface water, with the balance coming from groundwater [18,19].

### 2.2. Data Sources

2.2.1. Land Use Data Acquisition and Processing

We selected the Landsat scenes acquired for TRB in 1990, 1995, 2000, 2005, 2010 and 2015 epochs to reveal the changing characteristics of artificial oasis farmland area (http://www.gscloud.cn/, accessed on 10 April 2019). The remote sensing images were collected between August and September of the corresponding years to allow a more accurate vegetation interpretation due to cloud cover being at its lowest. The digital elevation model (DEM) data were from the ASTER GDEM V2 data in the geospatial data cloud with a nominal spatial resolution of 30 m. We have carried out large series of calibrations, such as geometric rough correction, radiometric calibration, geometric accurate correction,

atmospheric correction and removal of cloud (shadow) pixels, to avoid geometric or radiometric calibration errors. A remote-sensing classification software eCognition 8.7 was performed for the multiresolution segmentation and ground information extraction of the remote-sensing image. These temporally aggregated images were then combined with the high-resolution satellite data in Google Earth. Visual interpretation was performed with reference to stereoscopic aerial photographs and field surveys. It is estimated to have a high classification accuracy of 90–93%.

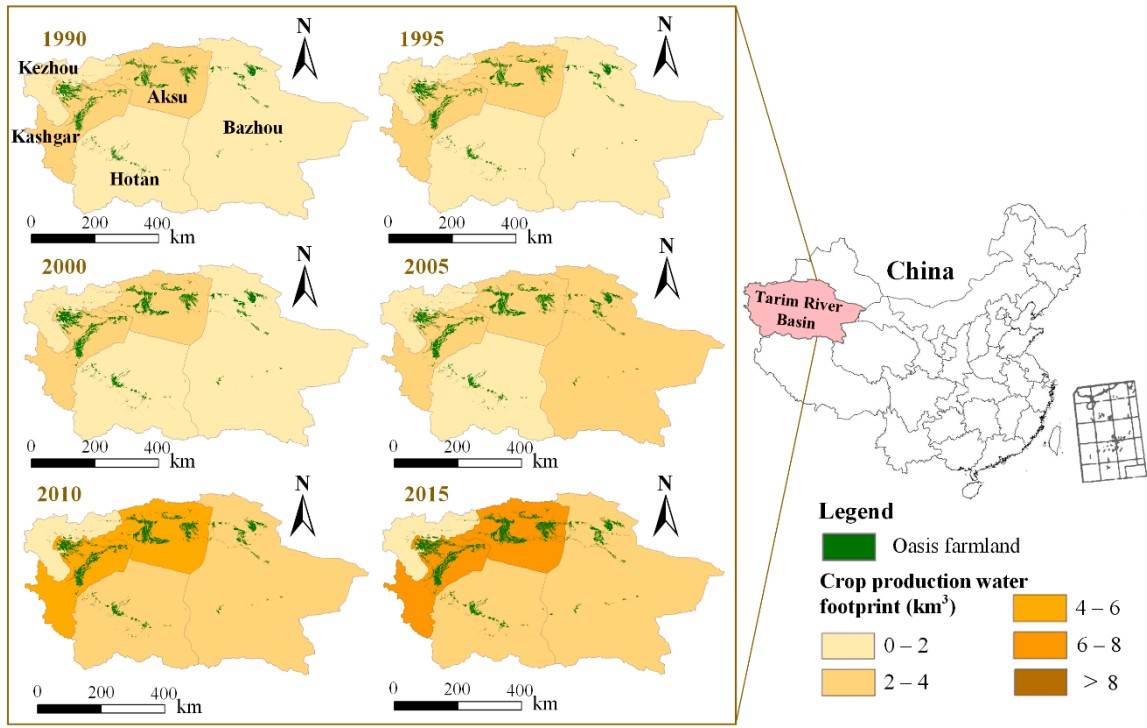

**Figure 1.** Location, crop production water footprint and oases of the study basin.

### 2.2.2. Meteorological, Water Resources and Statistical Data

Meteorological data (sunshine hours, wind speed, humidity, effective precipitation and air temperature) were obtained from 37 meteorological stations supplied by China Meteorological Data Sharing Service Network (http://data.cma.cn/, accessed on 10 May 2019). The Xinjiang Division of Water Resources is the provider of most of the data used for this study, such as municipal and industrial water resources development and utilization, agricultural and ecological water demand and water consumption. Additionally, the other statistical data were sourced directly from the Xinjiang Statistical Service, such as demographic information, crop yield, planting area and effective irrigation area.

### 2.3. Methodology

#### 2.3.1. Water Footprint of Crop Production

We calculated the production water footprint of 15 crops: rice, wheat, coarse cereals, soybeans, cotton, oil plants, sugar beets, vegetables, melons, potatoes, alfalfa, grapes, apples, fragrant pears and red jujube. These 15 crop types were estimated to occupy 97.4% of the total cultivated area in the TRB. Therefore, we took the sum of all these 15 crops as the TRB's crop production water footprint. First, we calculated the $ET_c$ (evapotranspiration) and $P_{eff}$ (effective rainfall) of these crops in each district during the growth period, respectively. The theoretical method commonly used to calculate $ET_c$ (mm) uses the Penman–Monteith method, which can accurately calculate $ET_c$ under different regions and climatic conditions [20]. $P_{eff}$ (mm) uses the CROPWAT model developed by the Food and

Agriculture Organization of the United Nations (FAO). Second, we calculated the $WF_{pro}$ (crop production water footprint), which can be divided into $WF_{blue}$ (blue water footprint) and $WF_{green}$ (green water footprint) from the water consumption perspective. The $WF_{pro}$ (m³), $WF_{blue}$ (m³) and $WF_{green}$ (m³) are expressed as follows [21]:

$$WF_{pro} = \sum_{n=1}^{n} WF_{i(blue,green)} \tag{1}$$

$$WF_{blue} = \left( 10 \times \sum_{d=1}^{\lg p} \max(0, ET_c - P_{eff})/Y \right) \times P_i \tag{2}$$

$$WF_{green} = \left( 10 \times \sum_{d=1}^{\lg p} \min(ET_c, P_{eff})/Y \right) \times P_i \tag{3}$$

In Equation (1) to Equation (3), $P_i$ is the total output of crop $i$ (t). $Y$ refers to the crop yield per unit area (t/ha). The factor 10 is the conversion coefficient of water depth into water per unit area of land area. $\sum$ is the cumulative amount of blue water (or green water) from planting period to harvest period. $\lg p$ is the length of the growth period (d).

### 2.3.2. Standard Deviation Ellipse Analysis

The Standard Deviation Ellipse (SDE), or directional distribution analysis, has unique advantages in exploring the spatial-temporal changes in resource factors. It can visually and accurately reveal these resource factors' evolution characteristics in two-dimensional space [22–25]. Therefore, by introducing the concept of the SDE, the spatial-temporal changes model of oasis agricultural water and land resources in TRB was constructed. The model judges the main trend direction and dispersion degree of the distribution of water and land resources in the oasis through the elliptical coverage area. We also calculated the moving direction and deviation distance of the center of gravity of $WF_{pro}$ and oasis farmland area ($OF_a$) and analyzed the change trajectories of their centers of gravity. If the study area consists of $n$ units and $(x_i, y_i)^t$ is the geometric coordinate of $i$-th unit ($i$ = 1, 2, 3, ..., $n$) at time $t$, the $WF_{pro}$'s gravity center coordinate G ($\bar{x}, \bar{y}$) at time $t$ can be written as:

$$G(\bar{x}, \bar{y}) = \left( \sum_{i=1}^{n} x_i \times m_i^t / \sum_{i=1}^{n} m_i^t, \ \sum_{i=1}^{n} y_i \times m_i^t / \sum_{i=1}^{n} m_i^t \right) \tag{4}$$

where $m_i^t$ is the attribute value of the $i$-th unit at time $t$.

The moving distance of the gravity center ($D$, km) can be calculated by:

$$D = C \times \sqrt{(\bar{x}_{t2} - \bar{x}_{t1})^2 + (\bar{y}_{t2} - \bar{y}_{t1})^2} \tag{5}$$

where $(\bar{x}_{t1}, \bar{y}_{t1})$ and $(\bar{x}_{t2}, \bar{y}_{t2})$ are the gravity center coordinate at the time $t_1$ and $t_2$ of the study and $C$ is a constant with the value of 111.111 (km), denoting the conversion coefficient from longitude and latitude coordinates on earth to plane distance.

The deviation angle ($\theta$) in the SDE model is the angle of clockwise rotation. The deviation angle ($\theta$), principal axis and auxiliary axis of the SDE can be calculated by:

$$\tan \theta = \left[ \left( \sum_{i=1}^{n} \widehat{x_i}^2 - \sum_{i=1}^{n} \widehat{y_i}^2 \right) + \sqrt{\left( \sum_{i=1}^{n} \widehat{x_i}^2 - \sum_{i=1}^{n} \widehat{y_i}^2 \right)^2 + 4(\sum_{i=1}^{n} \widehat{x_i y_i})^2} \right] / \left( 2 \sum_{i=1}^{n} \widehat{x_i y_i} \right) \tag{6}$$

$$\sigma_x = \sqrt{\sum_{i=1}^{n} (m_i^t \widehat{x_i} \cos \theta - m_i^t \widehat{y_i} \sin \theta)^2 / \sum_{i=1}^{n} (m_i^t)^2} \tag{7}$$

$$\sigma_y = \sqrt{\sum_{i=1}^{n} \left( m_i^t \widehat{x_i} \sin \theta - m_i^t \widehat{y_i} \cos \theta \right)^2 / \sum_{i=1}^{n} \left( m_i^t \right)^2} \tag{8}$$

where $\widehat{x_i}$ and $\widehat{y_i}$ are the deviations of gravity center and $\sigma_x$ and $\sigma_y$ are the standard deviations of principal axis and auxiliary axis, respectively.

### 2.3.3. Theory of Water Footprint and Farmland Matching

The Gini Coefficient (*GC*) proposed by economist Gini Corrado was first applied to the study of income disequilibrium based on the Lorentz curve [26]. Because the spatial distribution of natural resources shows diversity, the *GC* is also applicable in the study of the patterns of oasis water and land resource matching. To investigate the balanced state between $WF_{pro}$ and oasis farmland area ($OF_a$), the *GC* can be calculated as:

$$GC = 1 - \sum [l \times (2a - \omega)] \tag{9}$$

where $l$ and $\omega$ refer to, respectively, the proportions of $OF_a$ and $WF_{pro}$ in each district, and $a$ reference to the cumulative percentage of the $OF_a$ of the TRB. The *GC* ranges from 0 to 1. A greater degree of equilibrium between $WF_{pro}$ and $OF_a$ will lead to a smaller *GC* and vice versa.

Following the study of Gobillon et al. (2007) [27], spatial mismatch is used to describe the imbalance between water and land resources in space, which has a profound impact on regional water management and land-use policies. The spatial mismatch index was introduced to measure the spatial relationship between $WF_{pro}$ and $OF_a$. The formula is as follows:

$$SMI(WF_{pro}\_OF_a)_i = \left( \frac{WF_i}{\sum\limits_{i=1}^{n} WF_i} - \frac{OF_i}{\sum\limits_{i=1}^{n} OF_i} \right) \times 100 \tag{10}$$

$$\sum SMI(WF_{pro}\_OF_a) = \sum_{i=1}^{n} \left| SMI(WF_{pro}\_OF_a)_i \right| \tag{11}$$

where *SMI(WF$_{pro}$_OF$_a$)$_i$* refers to the spatial mismatch index of region $i$ between $WF_{pro}$ and $OF_a$, $WF_i$ and $OF_i$ refer to, respectively, the region $i$'s crop production water footprint (km$^3$) and oasis farmland area (km$^2$), and $\sum SMI(WF_{pro}\_OF_a)$ refers to the total level of spatial mismatch in TRB. A higher value of *SMI(WF$_{pro}$_OF$_a$)$_i$* means there is more $WF_{pro}$ in each of unit oasis farmland of region $i$, and a reduced *SMI(WF$_{pro}$_OF$_a$)$_i$* means the oasis farmland resource is utilized extensively and the $WF_{pro}$ is weakened in region $i$. In order to judge whether the spatial relationship between $WF_{pro}$ and $OF_a$ is matching in each region, we chose the Jenks Natural Breaks Classification to help us set standard values of the results [28]. If $\left| \sum SMI(WF_{pro}\_OF_a)_i \right|$ is less than the standard value, it shows that the relationship between $WF_{pro}$ and $OF_a$ is a spatial match in region $i$ and vice versa.

### 2.3.4. Sensitivity Index (*SI*)

A sensitivity assessment index (*SI*) was introduced to quantitatively analyze oasis farmland's sensitivity to $WF_{pro}$. It aims to explain the variations in $OF_a$ caused by the change in $WF_{pro}$ for investigating the potential impacts between them. The formula is as follows:

$$SI_i = \left| \frac{(OF_{t2} - OF_{t1})/OF_{t1}}{(WF_{t2} - WF_{t1})/WF_{t1}} \right| \tag{12}$$

where $SI_i$ represents the value of sensitivity index in region $i$, $OF_{t1}$ and $OF_{t2}$ are the $OF_a$ at the start and end of study period (km$^2$), and $WF_{t1}$ and $WF_{t2}$ refer to, respectively, $WF_{pro}$ at the beginning and end of study period (m$^3$). A higher value of $SI_i$ reflects a higher sensitivity of $OF_a$ to $WF_{pro}$ change, which means that a great change in $OF_a$ could be altered by the subtle change in $WF_{pro}$ [29].

## 3. Results

### 3.1. Crop Production Water Footprint and Farmland Resources

Figure 2 depicts the changing trends of the crop production water footprint ($WF_{pro}$) and oasis farmland area ($OF_a$) in the TRB. Results showed that the total $WF_{pro}$ increased by 184.2% ($WF_{blue}$ and $WF_{green}$ increased by 185.4% and 175.0%, respectively) in the past 25 years. The annual average value of $WF_{pro}$ showed obvious differences between districts; in descending order of this value, the disctricts were Kashgar, Aksu, Bazhou, Hotan and Kezhou. Among them, the $WF_{pro}$ of Kashgar was 14.96 times larger than that of Kezhou. Similarly to $WF_{pro}$, the $WF_{blue}$ of each district showed a progressively increasing trend from 1990 to 2015. Kashgar made the largest contribution (36.1%) to the components of the $WF_{blue}$ among all the districts in TRB. Aksu and Bazhou, ranked in second and third place, were 32.9% and 15.9%. Compared to the $WF_{blue}$, the inter-annual variability of the $WF_{green}$ showed a relatively slightly increasing trend in each district. The $WF_{pro}$ in each region was dominated by blue water (~87.3%), and the proportion of green water was relatively small (~12.7%).

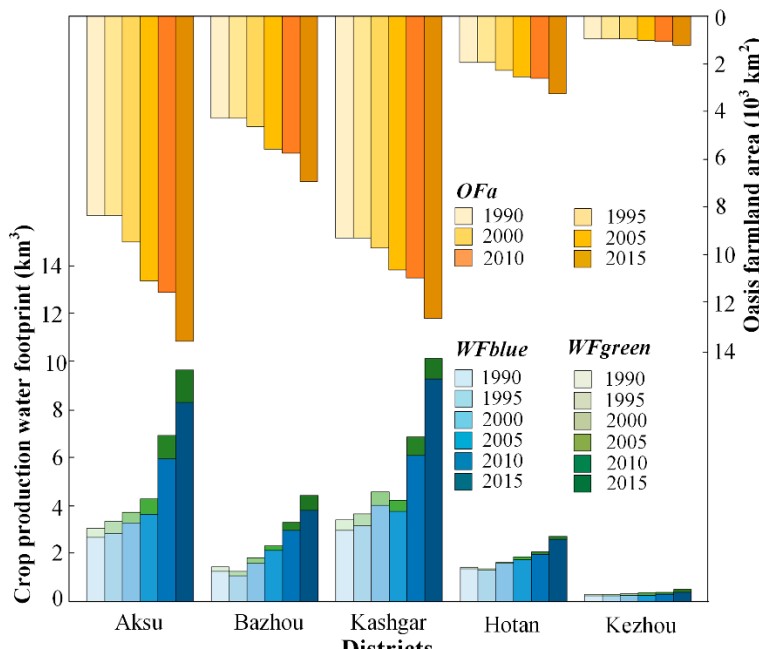

**Figure 2.** Recent trends in crop production water footprint and farmland area in each district of the Tarim River Basin (TRB).

$OF_a$ increased by 51.9% from 1990 to 2015. The inter-annual variability of each district had similar changing trends of moderately increasing during the entire study period. The annual average $OF_a$ ranking of the districts was in descending order in Kashgar, Aksu, Bazhou, Hotan and Kezhou. The sum of $OF_a$ in Kashgar and Aksu accounts for more than 70% of the $OF_a$ in TRB. Overall, it indicated that Kashgar and Aksu constitute the two main districts of the $WF_{pro}$ and $OF_a$ development in the TRB.

### 3.2. Spatial Distribution Characteristics of Crop Production Water Footprint and Oasis Farmland

As shown in Figure 3, the spatial distribution of $WF_{pro}$ and $OF_a$ in TRB is not uniform and presents the characteristics of large changes of $WF_{pro}$ and $OF_a$ in Aksu and Kashgar, and a smaller change of $WF_{pro}$ and $OF_a$ in other districts. The change in $WF_{pro}$ and $OF_a$ of some districts show a degree of mismatch. For example, the changes in $WF_{pro}$ in Aksu and Kashgar were roughly consistent (both greater than 5 km$^3$), but the change in $OF_a$ in Kashgar was only 63.9% of that of Aksu.

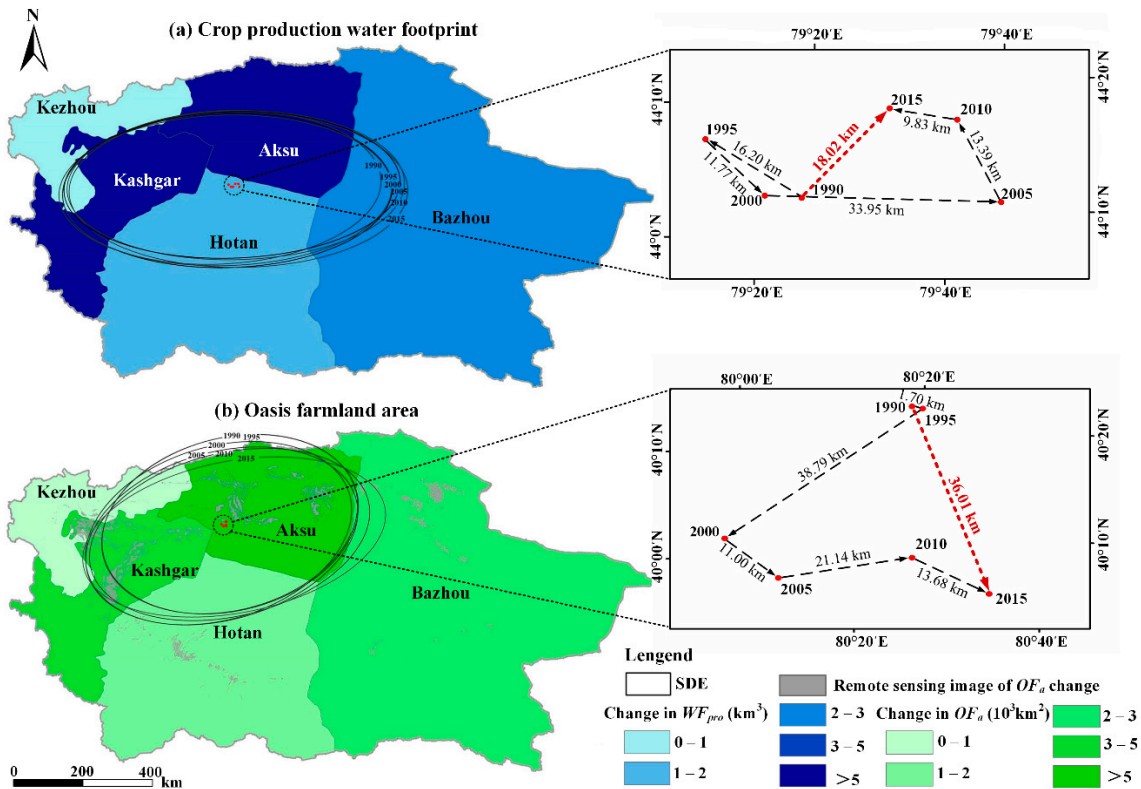

**Figure 3.** Change and migration trajectories of crop production water footprint *(WF_pro)* and oasis farmland area (*OF_a*) in TRB during 1990 to 2015.

In the past 25 years, the movement pattern of $WF_{pro}$'s center of gravity was roughly inconsistent with that of $OF_a$. The gravity center of $WF_{pro}$ moved more obviously in longitude. In terms of moving direction, the gravity centers of $WF_{pro}$ and $OF_a$ both moved to the southeast between 2000 and 2005 but were different in other periods. In terms of moving distance, the location characteristics of $OF_a$ gradually expanded 36.01 km to the southeast, while the center of gravity of $WF_{pro}$ moved 18.02 km to the northeast during the entire period. The centers of gravity of $WF_{pro}$ and $OF_a$ both moved to the east, showing that $WF_{pro}$ and $OF_a$ in eastern regions accounted for the increasing proportion in the whole study area.

### 3.3. Spatial Matching Analysis of Crop Production Water Footprint and Oasis Farmland

The results of spatial mismatch index (*SMI*) and Gini coefficient (*GC*) in TRB are shown in Figure 4. From 1990 to 2005, the *SMI* ($WF_{green}\_OF_a$) presents an increasing trend, while the *SMI* ($WF_{blue}\_OF_a$) presented a roughly decreasing trend. In the same period, the distance between the centers of gravity of $WF_{pro}$ and $OF_a$ was closer, as seen in Figure 3. These showed that the spatial mismatch between $WF_{pro}$ and $OF_a$ was remedied because agricultural water was dominated by $WF_{blue}$ in TRB. From 2005 to 2010, both the *SMI* and *GC* of ($WF_{blue}\_OF_a$) presented a decreasing trend, and the spatial match between $WF_{blue}$ and $OF_a$ was improved again. The *GC* ($WF_{blue}\_OF_a$) was on a decline after 2010, but the *SMI* sharply increased from 8.34 to 13.09, which reflected that the degree of equilibrium between $WF_{blue}$ and $OF_a$ decreased. On the whole, the variation tendency of the *SMI* and *GC* reveal that the spatial mismatch between $WF_{blue}$ and $OF_a$ in TRB was gradually remedied after 2000. However, the spatial mismatch between $WF_{green}$ and $OF_a$ was gradually exacerbated.

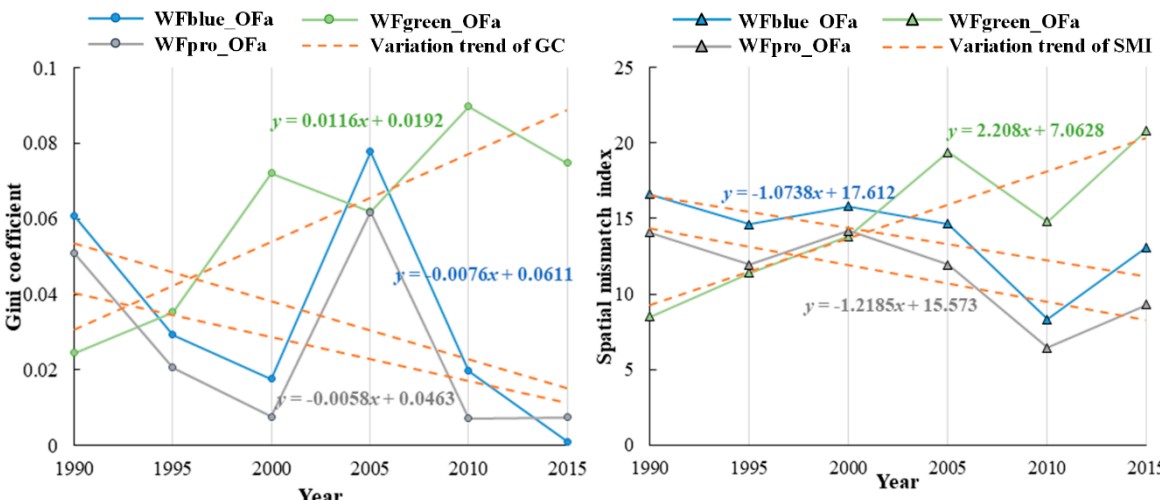

**Figure 4.** The Gini Coefficient and spatial mismatch index of crop water footprint and oasis farmland.

Using ArcGIS 10.2, we set −2.0 and 2.0 as the standard values to visualize the matching patterns between $WF_{pro}$ and $OF_a$ at each district based on Natural Breaks Classification (Jenks), as shown in Figure 5. From 1990 to 2015, the $OF_a$ with serious spatial mismatch and inefficient use of blue water was mainly distributed in Hotan, and the $OF_a$ with serious spatial mismatch and inefficient use of green water was mainly distributed in Aksu. From 2010 to 2015, the $OF_a$ with inefficient blue water use extended to Kashgar. The $OF_a$ with highly efficient blue water utilization was mainly distributed in Aksu and Bazhou. From 2000 to 2010, The $SMI$ ($WF_{blue}\_OF_a$) of Aksu and Bazhou started to decrease, coupled with the increase in $SMI$ ($WF_{green}\_OF_a$). As can be seen in Figure 3, the distance between the center of gravity of $WF_{pro}$ and that of $OF_a$ was constantly getting closer, indicating that agricultural water use efficiency of Aksu and Bazhou improved. According to the results shown in Figures 3 and 4, we also found that the improvement of Hotan with agricultural water use efficiency was the reason why the $SMI$ ($WF_{pro}\_OF_a$) and $GC$ ($WF_{pro}\_OF_a$) both decreased and the spatial mismatch in Hotan decreased from 2000 to 2010. Until 2015, the poor spatial mismatch in Hotan was remedied due to the relative stability of increasing trends in the $WF_{pro}$ and agricultural water use efficiency.

### 3.4. Sensitivity Analysis of Crop Production Water Footprint and Oasis Farmland

We calculated the sensitivity index ($SI$) of $OF_a$ to $WF_{pro}$ of each district in TRB, as shown in Figure 6. The sensitivity was divided into four levels: 0< $|SI| \leq 0.5$ indicates that $OF_a$ is non-sensitivity to changes in $WF_{pro}$, $0.5 < |SI| \leq 1.0$ indicates low sensitivity, $1.0 < |SI| \leq 1.5$ indicates moderate sensitivity, and $|SI| > 1.5$ indicates high sensitivity.

The number of districts with moderate and high sensitivity of $WF_{blue}\_OF_a$ first increased and then decreased, while the quantity of districts with non- and low sensitivity first decreased and then increased during the study period. During 2000–2005, there were four districts with moderate and high sensitivity. In comparison, all districts in other periods had non- and low sensitivity. Especially in 1990–1995 and 2005–2010, the $OF_a$ in all districts was non-sensitive to $WF_{blue}$ change in TRB, while only two districts in TRB showed non-sensitivity during 2010–2015 (Kashgar and Aksu). In addition, during the study period, there were no districts with the high sensitivity of $OF_a$ to green water footprint change in TRB. During 1995–2000 and 2000–2005, there was one moderate-sensitive area in each. During 2010–2015, the number of mid-sensitive areas rose to two. The $OF_a$ in all districts were mainly non-sensitive or low-sensitive to green water footprint change during other periods. From the perspective of spatial distribution, the districts with moderate and high sensitivity of $OF_a$ to the $WF_{blue}$ in TRB were gradually expanded westward. Especially during 2000–2005, the $OF_a$ in Aksu and Kashgar, located in the northwest of the TRB, was

significantly more sensitive to $WF_{blue}$ change than the other three districts. However, the sensitivity of $OF_a$ to $WF_{green}$ change was relatively insignificant in spatial distribution.

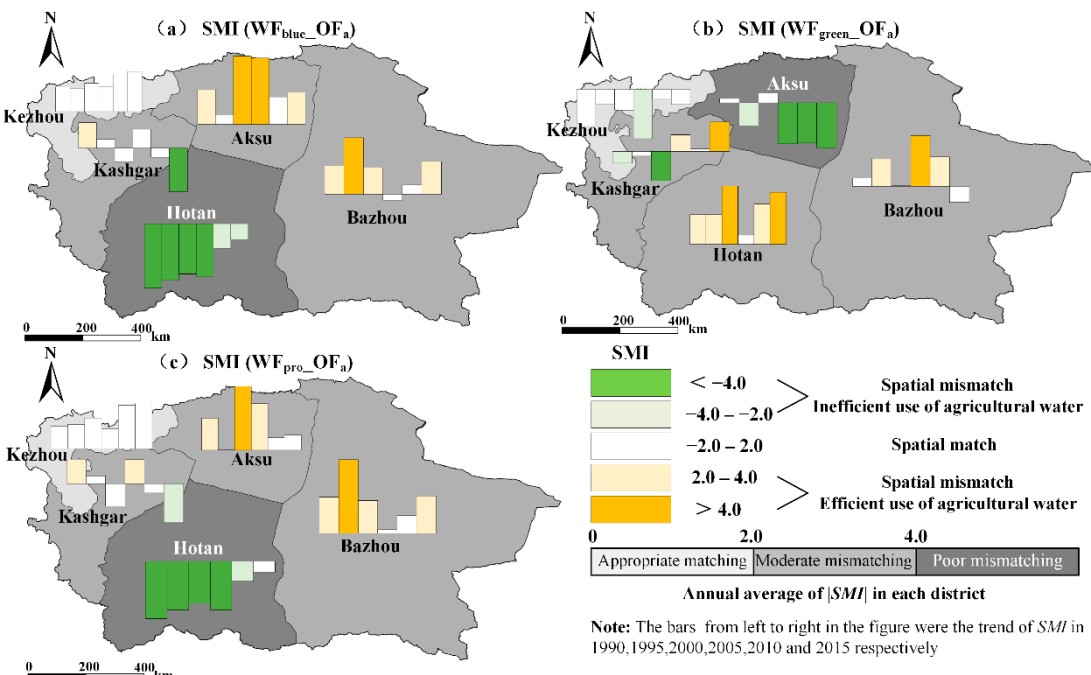

**Figure 5.** The spatial-temporal pattern of spatial mismatch index from 1990 to 2015. (**a**) The spatial mismatch index between blue water footprint and oasis farmland area (*SMI (WF$_{blue}$_OF$_a$)*); (**b**) The spatial mismatch index between green water footprint and oasis farmland area (*SMI (WF$_{green}$_OF$_a$)*); (**c**) The spatial mismatch index between crop production water footprint and oasis farmland area (*SMI (WF$_{pro}$_OF$_a$)*).

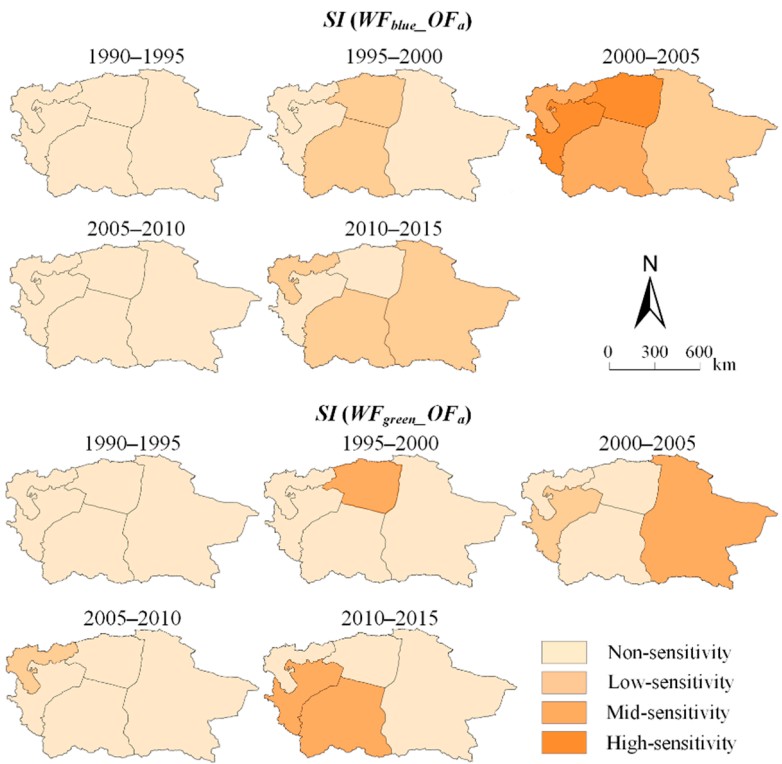

**Figure 6.** The spatial-temporal pattern of sensitivity from 1990 to 2015.

## 4. Discussion

### 4.1. Main Factors Influencing Crop Water Footprint and Farmland Resource Changes

Both the $WF_{blue}$ and $OF_a$ continue to increase and aggregate in spatial-temporal changes based on the historical trends. Without the implementation of the corresponding regulatory policies, the growth of $WF_{blue}$ driven by the expansion of $OF_a$ will lead to a reduction in ecological land, which will result in obvious declines in habitat quality, water economic production efficiency and carbon sequestration [30]. As the marginal efficiency of external inputs factors (urbanization and agricultural mechanization) has declined, the agricultural production scale has continued to expand in TRB. In future sustainable development scenarios, it is recommended that the government proposes a plan to avoid long-term expansion of oasis farmland, which is the most effective in maintaining the suitable oasis farmland scale. Under this policy scenario, both $WF_{blue}$ and $WF_{green}$ will decline. In addition, future sustainable development policies should also effectively maintain and improve carbon sequestration and habitat quality by reducing the occupation of ecological land by farmland. Existing studies have shown that the implementation of such management measures can enhance the overall ecosystem service value [31,32]. The development of water-saving technology has proven to be highly effective in improving water efficiency for policymakers. The water-saving irrigated area in the TRB has increased by 7033 km$^2$ and is irrigated by drip irrigation systems covering the period between 2005 and 2015. With the widespread adoption of water-saving engineering technologies such as prevention of channel leakage, water delivery with low-pressure pipe and drip irrigation systems, the $WF_{pro}$ per unit yield of some crops in the TRB has declined [33]. However, water-saving irrigation scheduling, optimized water allocation in irrigation areas and irrigation water demand forecasting technology have not been improved. These can be the key drivers of increasing local $WF_{blue}$. The global green plus blue global WF benchmark methodology has been widely used when assessing agricultural water efficiency [34]. Nevertheless, the TRB has a typical temperate continental arid climate, characterized by low rainfall, high evapotranspiration and extremely scarce green water resources. Therefore, the TRB dominated by irrigated agriculture has lower green water production efficiency as compared to the humid regions dominated by rain-fed agriculture [35]. More attention should be paid to the relationship between agricultural production and ecological restoration during the implementation of future development policies, developing better environmental flows assessment tools to accurately estimate the extent of water scarcity per catchment and to improve integrated water and land resources management level [36,37].

In terms of improving water efficiency, we suggest that more attention should be paid to the relationship between agricultural production and ecological restoration, and to improving integrated water and land resources management levels during the implementation of future development policies. This can ensure reasonable land utilization and development under the constraints of the regional water resource carrying capacity and can reduce the pressure caused by excessive $WF_{pro}$ while satisfying agricultural development. $OF_a$ was shown to be significantly affected by the climatic characteristics and water quality conditions in Hotan and Aksu. Moreover, the change in land use transition and crop plant structure can also change the $OF_a$ in the two districts by affecting the $WF_{pro}$ [38]. Due to the accelerated process of urbanization, a large amount of oasis farmland was inefficiently utilized, abandoned or transferred into other land types [39]. Farmers in Aksu and Bazhou interested only in profit potential would therefore be more inclined to plant economic crops (e.g., cotton and red jujube) rather than food crops to keep their income and living standard more stable. The large-scale production of these high water-intensive economic crops have further increased the local $WF_{pro}$ per unit yield.

### 4.2. Implications and Suggestions from Spatial Matching Theory and Sensitivity Analysis

Although the relationship between irrigation water and effective irrigated area distribution patterns has been noted by a number of research studies, the correlation between the actual water use of crops and farmland landscapes has not yet been fully appreci-

ated [40]. It was believed that the amount of water resource consumed by humans in the entire economic production process (including physical water and virtual water) and the scale of farmland were more closely related in time and space. In the present study, the $WF_{pro}$ and $OF_a$ of TRB were moving from the northern regions with a scarcity of water resource conditions to eastern regions, while the economic water productivity per unit area of crops in the eastern regions did not increase significantly. In the long run, this inconsistent distribution pattern will pose risks in terms of local agricultural production. In TRB, the phenomenon of spatial mismatch between $WF_{pro}$ and $OF_a$ is more obvious, which can cause problems such as an imbalance in regional water and land structures and ecological risks. Meanwhile, the increase in the number of moderate- and high-sensitivity districts also revealed that the $WF_{pro}$ has a greater impact on $OF_a$ changes. However, with the scarcity of high-quality farmland and the rapid development of the primary industry, it has become increasingly difficult to implement Returning Farmland to Forest Program (RFFP) by controlling the $WF_{blue}$ [41].

The Chinese government has implemented a series of policy transmission mechanisms to rationally allocate water and farmland resources. These decision-making sessions have proven effective and feasible in some pilot regions with water shortages [42]. Measures such as changing the farmland operation scale and promoting farmland transfer are based on the successful experience of the above-mentioned pilot regions. Briefly, the TRB's policy-makers can try to take these measures to alleviate the structural imbalance between $WF_{blue}$ and $OF_a$. Policymakers can use subsidy policies to realize such a transformation of TRB from small-scale to moderate-scale agricultural production. Promoting the farmland transfer may enhance the contribution of $WF_{pro}$ per unit to the improvement of crop productivity and benefit to some extent, including subsidizing moderate-scale farmland management, improving the registration system of farmland contractual management rights, standardizing control of irrigation water use and encouraging innovation in forms of farmland transfer. The cross-regional water transfer and farmland scale control integrated planning should be carried out to alleviate the current disparities in agricultural development and water use. In oasis areas with vulnerable ecologies, crop rotation and fallow systems are gaining greater attention for sustainable intensification of agro-ecosystem. It is essential to encourage farmers to increase the proportion of low-water-intensive crops planted and render or increase subsidies for farmers. The role of markets should be given full play in the optimization of farmland resource allocation to conduct a pattern of Transferrable Development Rights transaction (TDR) [43]. Moreover, modern oasis land-use management practices may achieve a win–win effect between the improvement in blue water utilization efficiency and quality assurance of oasis farmland. $WF_{green}$ is a kind of water consumption restricted by the natural conditions of the region; unlike $WF_{blue}$, its impact on the change in $OF_a$ was limited. At present, the effective management of green water can only be achieved via virtual water trade [44]. The government should focus on China's "One Belt One Road" initiative so as to optimize and develop the product structure of processing and service industries. Compared with the agriculture sector, the direct blue–green physical water consumption and virtual water export volume per unit output in secondary and tertiary industries were relatively low [45].

## 5. Conclusions

Using TRB as a case study, this paper assessed the spatial-temporal variability of $WF_{pro}$ and $OF_a$ over the period 1990–2015. Furthermore, the potential impact of $WF_{pro}$ changes in $OF_a$ was examined, which facilitates an improved understanding regarding the blue–green water use efficiency.

On the whole, the $OF_a$ was more strongly associated with $WF_{blue}$ change than $WF_{green}$. In 1990–2015, the increasing trend of $WF_{blue}$ was more remarkable in various districts compared to that of $WF_{green}$. The average annual $WF_{blue}$ apparently grew more quickly than $WF_{green}$ in TRB. The remarkable, significant increase in the TRB's $WF_{blue}$ in the study period mainly occurred in Kashgar and Aksu districts, where the oasis farmland area

displays a continuous trend of expansion. The analysis changes in $WF_{pro}$ and $OF_a$ in the TRB demonstrated that the SDE model gives insights into spatial distribution characteristics analysis. Our findings show that the spatial aggregation of $WF_{pro}$ and $OF_a$ in the eastern districts of TRB continuously increased.

Assessing the spatial mismatch between $WF_{blue}$ and $OF_a$ in TRB, we found that the overall level of spatial mismatch fluctuated during the study period. After 2000, the spatial mismatch between $WF_{blue}$ and $OF_a$ in TRB was gradually remedied, but the spatial mismatch between $WF_{green}$ and $OF_a$ gradually deteriorated. The districts with the most inefficient use of blue water were mainly distributed in the oasis farmland of Hotan. With the popularization of water-saving technologies in oasis farmland areas of Aksu and Bazhou, these two districts became more dominant over time in the regional pattern of crop production and agricultural security in the TRB.

The spatial-temporal pattern of sensitivity showed that the number of districts with moderate and high sensitivity to changes in the $WF_{blue}$ increased continuously and the spatial distribution of districts shifted from the east to the west during 1990–2005. The change in $WF_{blue}$ has a great increasing influence on $OF_a$ in TRB with time, especially in the western districts. The continuous acceleration of urbanization and the excessive reclamation of farmland after 2005 led to the inefficient use of a large amount of oasis farmland, which lowered the $SI$ ($WF_{blue\_}OF_a$) in the TRB. For the sustainability of the oasis agricultural economy, TRB's government should improve the local agricultural blue–green water use efficiency by changing the oasis farmland operation pattern and promoting farmland transfer. These ways can not only increase the industrial diversity of agriculture but also have great importance for controlling oasis farmland scale.

**Author Contributions:** J.Y. and X.D. wrote the manuscript and prepared figures. A.L., X.H., H.G. and J.D. advised the study design and data analyses. J.Z. and C.R. collected the data. A.L. and X.D. revised the paper. All authors discussed the results and commented on the manuscript. All authors have read and agreed to the published version of the manuscript.

**Funding:** The authors recognize the research support of the Ministry of Science and Technology of the People's Republic of China through grants 2016FYA0601600 and 2017YFC0404300. This work was also supported by the National Natural Science Foundation of China (U1803244).

**Acknowledgments:** Conversations with Xu Zhao were helpful in this research. Comments from Pieter van Oel were critical in bringing important focus to this study. The authors thank Yawen Shao from the University of Melbourne for polishing the language of the manuscript. We are grateful for the time and valuable comments of the anonymous reviewers in improving this paper.

**Conflicts of Interest:** The authors declare no conflict of interest.

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
