# Peer review of "Understanding the Spatial-Temporal Changes of Oasis Farmland in the Tarim River Basin from the Perspective of Agricultural Water Footprint"

_water, doi:10.3390/w13050696_

Round 1

Reviewer 1 Report

Dear authors,

Thank you for your exploration of how we can better understand ineffective use of water resources from biological and management points of view. Here are some suggestions that should improve the quality of your paper:

  1. Abstract, line 24: The sentence 'The degree of spatial...' doesn't make sense if not deleted words 'extensively' and 'and' in line 25.
  2. Abstract, line 26: 'affected' instead of 'affect'.
  3. 2.3.1.Water footprint of Crop Production, line174: 'were estimated' instead of 'are estimated'.
  4. It is missing a more precise and more addressed authors' suggestion than it is in line 464-466.
  5. It is missing a more precise and more addressed authors' suggestion than it is in line 531-533. Since this paper is an input to a management, it is needed to say who will manage that prevention and in which way?
  6. In line 543 should be 'We are grateful...'

Reviewer 2 Report

It seems to me that the paper correctly is discussed development of agricultural area and water use in oasis basin. 

It is understood the importance of index of water foot print and extension of farmland. Then, as assessment tool for water use and agricultural area condition, this methodology is  useful in future. 

However, in the discussion and conclusion, readers can not see how sustainable scenario is available in future through this analysis. Authors mentioned importance of water saving technology development for water use efficiency. Please consider the relationship between technological development and water use efficiency. And how water footprint would be expected to improve. If these points are additionally stated in discussion part, this analysis would be more impact for readers.

Thanks

Reviewer 3 Report

The authors assess "spatial mismatch" between the water footprint (WF) of production and agricultural land use in oases in the Tarim river basin in China.

I have following general observations:

  • The manuscript is not concise, is not to the point and written in a rather chaotic way. You loose the interest of the reader by an overload of information, much of which is not usefull
  • The visual presentations in the manuscript are of poor quality. This should be improved
  • the spatial resolution of the WF production is not clear, nor is the spatial resolution of the oases analysis - which seems to be based on satellite images - clear

For further consideration, the authors should:

  • make their manuscript concise. In total at least 1 third of the length of the manuscript can be reduced
  • figure 1 is useless, delete
  • Add a new Figure 1 which clearly shows the water footprint of production in the basin for the years you assess, and this in the highest spatial resolution in which you conduct your analysis. Add in this figure 1 in highest spatial resolution the location of your oases for the different years
  • clarify for which crops you calculate the WF of production. Is the WF of production you analyse the sum of all these crops?
  • check your English grammar throughout the manuscript, you make a lot of mistakes and typos
  • main element is that you should make your manuscript concise and to the point. Delete any unnecessary information
  • Clarify in the discussion very briefly what is the use of your analysis. Reduce the text of your discussion by at least 50%

Round 2

Reviewer 3 Report

The authors have improved their manuscript according to the recommendations of the reviewer(s). They have reduced the amount of text and increased the quality of their figures and overall presentation.

i have one remaining comment, regarding the (relatively newly written) section 4.1 "Main Factors Influencing Crop Water Footprint and Farmland Resource Changes":

The authors discuss water efficiency merely in the sense of blue water efficiency, e.g. by describing technical measures such as drip irrigation or prevention of channel leakage. They do mention integrated land and water management, but do not explicitely discuss the role of green water in this. Please add some sentences on "overall water productivity", including both blue and green water. Integrated land and water management also includes the overall blue+green WF productivity of agricultural production. You can mention 2 relevant papers for this: one on WF benchmarks - https://doi.org/10.1016/j.ecolind.2014.06.013 - and another newly published relevant paper https://doi.org/10.1016/j.scitotenv.2020.143992

The authors also briefly touch in this section on "the relationship between agricultural production and ecological restoration". They however do not mention "environmental flows". Please add 1 or 2 sentences on the important aspect of restoring and maintaining environmental flows and refer to the publications https://doi.org/10.1016/j.scitotenv.2017.09.056 as well as https://doi.org/10.1126/sciadv.1500323 

Round 3

Reviewer 3 Report

the authors have addressed the comments so I recommend for publication